# Evaluation of the performance of the Influenza-like Illness (ILI) surveillance system in the Okai Koi North District, Greater Accra Region, 2022

Doris Aboagyewaa Edu-Quansah[1]*, Delia Akosua Bandoh[1], Elijah Paa Edu- Quansah[2], Anthony Baffour Appiah[1], Charles Lwanga Noora[1], Ivy Asantewaa Asante[3], Dennis Laryea[4], Ernest Kenu[1]

1 Ghana Field Epidemiology and Laboratory Training Program, School of Public Health, Legon, Ghana, 2 African Field Epidemiology Network, Uganda, 3 Noguchi Memorial Institute for Medical Research, Legon, Ghana, 4 Disease Surveillance Department, Ghana Health Service

* dorisduke23@gmail.com

## Abstract

### Background

Influenza-like Illness (ILI) caused by the influenza virus, causes morbidity in Ghana. Records of ILI outbreaks in recent times and the COVID-19 pandemic disrupted surveillance activities, raises the quest to evaluate the ILI surveillance system. We evaluated the ILI surveillance system of Okai Koi North District to assess the system performance.

### Materials and methods

We adapted the CDC Updated guidelines for evaluating public health surveillance systems for this evaluation at Okai Koi North District. We extracted and reviewed ILI 2018-2021 morbidity data from the district's sentinel site and the National Influenza Center (NIC). We observed surveillance activities and interviewed key informants using an observational checklist and semi-structured questionnaire. Data were analyzed for frequencies and proportions, and results presented in charts and tables,

### Results

Of the 525 suspected samples reported from the district's sentinel site, 58 (11%) of 525 were Influenza positive with PVP, 11%. The system detected outbreaks over the evaluation period and has a year-round case detection. The system requires PCR for the detection of influenza virus. Nine (70%) of 13 staff indicated ILI surveillance system served as the backbone for case identification during the COVID-19 pandemic period. There is (89%) data completeness among sampled forms and data from the district and National Influenza Center. The system relies mostly on international

**Data availability statement:** All relevant data are within the manuscript and its Supporting Information files and other files.

**Funding:** The author(s) received no specific funding for this work.

**Competing interests:** The authors have declared that no competing interests exist.

donors. Nine (64%) of 13 staff confirmed no budget allocation for system operation. About 58% (306/525) of the ILI samples were transported to NIC for confirmatory test within the set 48 hours timeline.

## Conclusion

The system is useful as its meeting most of its objectives and it is sensitive. The system though complex, is flexible, representative, timely in most activities and has good data quality. We urge national stakeholders to establish thresholds for system.

## Introduction

Influenza poses a serious risk to public health worldwide, as both seasonal and pandemic strains have a significant negative influence on people's health, the economy, and society [1]. Influenza-like illnesses (ILI), also known as acute respiratory infection or flu-like syndrome, are caused by influenza viruses. They are infections that affect both the upper and lower respiratory systems and have signs and symptoms that are similar to influenza. The virus spreads directly from person to persons via aerosols and indirectly when person comes into contact with contaminated fomites [2]. The virus is self-limiting with an incubation period ranging from 1–4 days but can usually last for a week [3].

Due to the nature of the virus, its ability to pose health threats to individuals worldwide, coupled with epidemics in the past, the World Health Organization founded a surveillance network for influenza and other respiratory diseases, called Global Influenza Surveillance and Response System (GISRS). This system has been in existence since 1952 and its goal is to track the frequent alterations in influenza viruses and use vaccines containing strains that are currently in circulation to lessen the impact of influenza disease, with laboratory testing as its foundation [4]. Through GISRS, networking countries exchange clinical specimens and virus samples for genetic analysis throughout the year, via a global web-based tool called FluNet. This helps to better understand patterns of respiratory virus circulation, identify and characterize novel influenza viruses and other emerging respiratory viruses, and inform the development of seasonal influenza vaccine compositions.

Annually, influenza-like infections cause significant morbidity and mortality worldwide, particularly among children, pregnant women and the elderly. Extreme age increases the likelihood of complications, hospitalization, and influenza-related deaths significantly, particularly in older adults with underlying medical problems like diabetes or chronic heart disease [5,6]. Among the influenza subtypes linked to outbreaks include H1N1, H5N1, H7N9, H7N7, and H3N2.

Globally, seasonal influenza epidemics causes about 3–5 million severe illnesses and up to 650,000 respiratory fatalities. Its global attack rate is anticipated at 5–10% in adults and 20–30% in children [3]. Between 2010 and 2022, the United States Centers for Disease Control and Prevention (US CDC) also estimated that influenza caused about 9.4 million to 41 million illnesses, with 100,000 to 710,000

hospitalizations, and 4,900 to 52,000 deaths [7]. The WHO AFRO influenza laboratory network detected 8.4% (27,840) influenza positivity rate from 33,1369 specimen tested between 2018–2021 in the Africa region (FluNet Interactive Report, 2022). Between 2018–2021, the National Influenza Center (NIC) in Ghana tested a total of 23,527 specimen out of which 16.5% (3887) were influenza virus positive (FluNet Interactive Report, 2022).

In 2006, following the Avian influenza outbreak in 2005, Ghana, through the NIC commenced influenza surveillance in five health facilities within the affected regions. Objectives of the surveillance system included, obtain current information about transmission patterns, monitor circulating influenza strains, and better understand the epidemiology of influenza in Ghana. Surveillance for ILI increased steadily to about 25 sentinel sites prior to the COVID-19 pandemic to 35 sites in the country. The core mandate of the ILI surveillance system is to conduct testing of influenza and other respiratory viruses in Ghana, provide weekly virological reports to the Ghana Health Services (GHS) and through FluNet to GISRS.

Ghana in recent times, has recorded ILI outbreaks mostly in senior high schools as well as surge in ILI cases in a retrospective study [8–10]. These Flu outbreaks usually meet the lower levels of surveillance, example the district levels unprepared to mitigate them which either goes undetected or delayed detection. The Flu situation is not the same for all diseases, as some diseases under surveillance here in Ghana, example Measles have epidemic thresholds set for them. Thresholds denote a degree of incidence that is essential for timely detection of disease outbreaks, and its response [11]. In spite of these sporadic ILI outbreaks, the ILI surveillance system in Ghana was disrupted by COVID-19 from 2020 through to 2022 which resulted in the NIC focusing its testing only for COVID-19. However, following the drop in cases, COVID-19 was integrated into the ILI surveillance system in 2021. The integration concurrently revealed ILI outbreaks (including H3N2 and H1N1) in other parts of the country and the focal district. Okai Koi North District, the focal district of evaluation has recorded sporadic outbreak of influenza disease particularly, in some cluster of schools in the district, and it is known to have frequent influenza outbreaks [10]. An unpublished data from the Districts Health Directorate (DHD)-Ghana Health Service (GHS) in 2021, revealed an 18% Flu positive in the district. The disruption in surveillance activities, flu burden and outbreaks reported in the district, raises a quest to evaluate the ILI surveillance system. We present here findings on our evaluation of the ILI surveillance system in Okai Koi North District in assessing the systems performance in concert with its usefulness and system attributes.

## Materials and methods

### Evaluation design

We evaluated the Influenza-like illness (ILI) surveillance system in the Okai Koi North District, Greater Accra Region in June 2022 for the period 2018 to 2021. Major stakeholders at the national, regional, district and health facility levels were interviewed on their know-how about the surveillance system and their processes inputted into the system. A structured questionnaire developed based on the US CDC updated guidelines for evaluating surveillance systems was used for interviews and to garner data on the system usefulness and system attributes. We visited the major ILI sentinel site, which is in the district, to assess epidemiological and laboratory data (including sample collection, storage, transportation, data reception, analysis and dissemination) on ILI cases while observing surveillance activities as well. Other healthcare facilities (non-sentinel sites) visited focused on epidemiological data only, entered the District Health Information and Management System 2 (DHIMS 2) as well as their routine records. We abstracted and reviewed routine ILI data (between January 2018 to December 2021) from the sentinel site as well as the National Influenza Center (NIC) on January 6th, 2022. The abstracted data were declassified or without personal identifiers.

### Evaluation area

The retrospective evaluation process of the ILI surveillance system was done at all surveillance levels (national, regional, district and subdistrict), with focus in the Okai Koi North District. Okai Koi North District is a newly created district, inaugurated in 2018 and it is one of the 26 metropolitan, municipal and district assemblies (MMDAs) in the Greater Accra Region,

Ghana. It is an urban settlement, densely populated with an estimated population size of 296,742. It has a land area of 21.34km². Okai Koi North District has 3 subdistricts namely: Achimota, Abeka and Akweteyman, with Achimota being its capital. It has 17 healthcare facilities; 14 private and three public, with 22 functional Community Health Based Planning Services (CHPS) zones bordering around the only forest reserve, Achimota forest, in the capital city. There are cluster of schools with boarding facilities in the district where most of the students travel from all over the country for school. The district is served mostly by the Achimota Government Hospital, which serves as the only influenza sentinel site, and works in collaboration with the NIC. Fig 1. Below represents health facilities or areas under Okai Koi North District and other institutions visited for the conduction of the ILI surveillance evaluation.

## Evaluation approach and data collection technique

Adapting the CDC's Updated Technical Guidelines for evaluating public health surveillance systems, a semi-structured questionnaire and a checklist were developed for evaluating the ILI surveillance system's usefulness and system attributes which included simplicity, flexibility, data quality, acceptability, sensitivity, predictive value positive, representativeness, timeliness, and stability [12]. We selected the Flu sentinel site for this evaluation, the DHD and then four non-sentinel sites were randomly sampled for the evaluation. The non-sentinel sites were evaluated to understand their involvement and contribution towards ILI surveillance and to know whether they report to the next surveillance level.

At the sentinel site and the main DHD, three key persons involved in surveillance were randomly selected and interviewed using the developed questionnaire. Whereas, at the non- sentinel sites visited, two key persons were randomly selected and interviewed, however, at sites where persons involved in the ILI system are limited, one person was selected

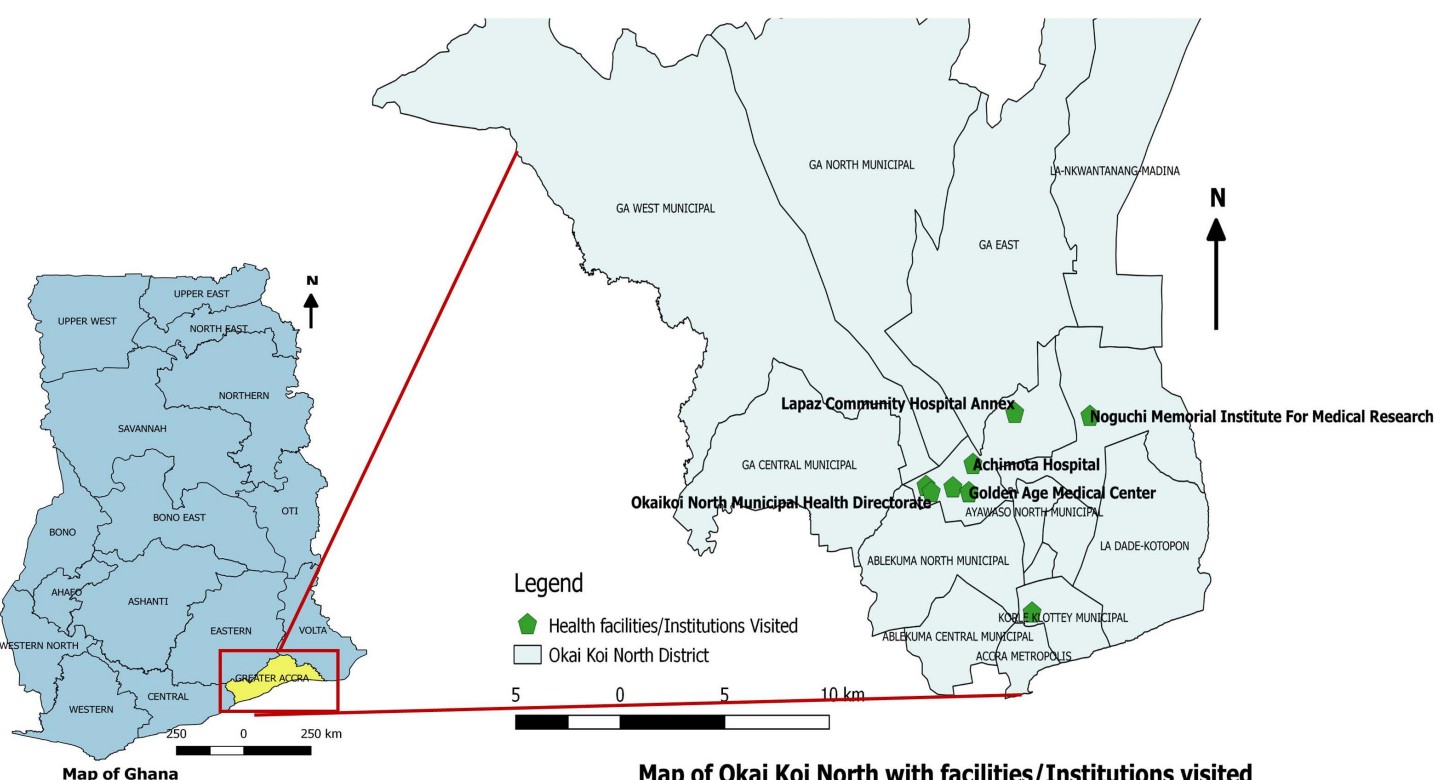

**Fig 1. Map of health facilities and institutions visited for the ILI surveillance system evaluation, Okai Koi North District, 2022.**

for the interview. The questionnaire included indicators such as sample collection, storage, transportation and/ data reception, analysis, and findings dissemination.

Hard copies of filled-out case-based forms or reporting forms within the evaluation period from the sentinel site were checked for availability. About 10% of these forms were randomly sampled and reviewed for completeness and accuracy, and these were validated with data from the NIC.

Surveillance data from the NIC, which is data from the sentinel site and DHIMS 2 were reviewed and extracted using the developed checklist. Also, we reviewed consulting room registers and OPD records from all the health facilities visited within the evaluation period, using the developed checklist. The evaluation targeted information from records (data and forms) on all in-patients and out-patients, males and females reporting at these visited health facilities in the district as well as data on samples received at the NIC. Similarly, surveillance activities were also observed using the checklists. The purpose of the observation carried out related stakeholders' answers to questionnaire to what was practiced on the field.

## Approach to evaluating usefulness and attributes

Adapting the US CDC updated guidelines for evaluating public health surveillance systems, we assessed the system using the underlisted indicators and their assessment levels, as shown in Table 1.

## Data analysis

Data from the paper-based questionnaire were entered into Excel 2015, cleaned and imported into Epi-Info version 7. Key indicators set by the surveillance system was analyzed descriptively from the quantitative data garnered and presented as frequencies, proportions, charts, and tables. We generated spot map indicating the locations of the sentinel sites and non-sentinel sites visited within Okai Koi North District, using QGIS desktop, 2.14.3. We generated an ILI epidemic threshold for Okai Koi District from data garnered from the NIC using the CUSUM - C2 method. The four years data were arranged by year and months (January – December). For each threshold point, we took the mean and standard deviation for a group of seven surveillance points (months over the four years) in the past, skipping over the two most recent points (months). We then added three times the standard deviation to the mean value to obtain the C2 value for specific point/month [C2 = Mean + 3(standard deviation)]. The calculated C2 values for the months were used to plot the epidemic threshold chart using Microsoft Excel.

## Ethical consideration

Ethical Review Committees were not required to formally review the evaluation because it was done within the matrix of Integrated Disease Surveillance and Response by the Ghana Health Service and as part of the mandate of the Disease Surveillance Department of the Health Service. Permission was however obtained from the Ghana Health Service's Public Health Directorate, the NIC and the authorities of the health facilities visited. The evaluation was also approved by the Ghana Field Epidemiology and Laboratory Training Programme (GFELTP).

## Results

### Background or description of respondents

A total of 13 respondents from the four surveillance levels (national, regional, district and subdistrict level) participated in the ILI surveillance system evaluation process. Majority, 54% (7/13) of the respondents were from the six facilities (i.e., DHD, both sentinel and non-sentinel sites) visited in the district. The predominant cadre amongst the staff interviewed were data managers, 23.1% (3/13). The remaining were surveillance officers, Epidemiologist, laboratory person and clinician. Of these, 85%, (10/13) had received Integrated Disease Surveillance and Response (IDSR) training and same proportion knew the ILI case definition and the component of the system.

**Table 1. Indicating assessment or grading usefulness and system attributes for evaluating the ILI surveillance system at Okai Koi North District, Accra-Ghana, 2022.**

| Usefulness and Attributes | Indicators Assessed | Expected Response | Grading or Assessment |
|---|---|---|---|
| **Usefulness** | - Timely detection of diseases and its trends (using thresholds)<br>- Characterize circulating virus<br>- Estimate mortality and morbidity<br>- Detect epidemics | Yes<br>Yes<br>Yes<br>Yes | - Useful = Yes to ≥ 3 questions<br>- Not useful = Yes to < 2 questions |
| **Simplicity** | - Case-based form easy to fill<br>- Case definition easily understood and applied by staff<br>- Required laboratory for testing and confirmation | Yes<br>Yes<br>Yes | - Simple = Yes to case-based forms easy to fill, easily understood and applied by staff – Responses to last – Complex = Yes to system requires laboratory for testing and confirmation |
| **PVP** | - Proportion (%) of suspected cases confirmed by lab (NIC | – | $PVP = \frac{Number\ of\ laboratory\ confirmed\ ILI\ cases}{Total\ number\ of\ suspected\ cases\ tested} \times 100$ |
| **Representativeness** | - Accurately represent population (sex and age)<br>- Includes both in-patients and out-patients<br>- Health facilities (both public and private facility) reporting respiratory symptoms to the district level | Yes<br>Yes<br>Yes | - Representative = Yes to;<br>1. System covers persons of both sexes and all age groups<br>2. System covers both in-patients and out-patients (enrolls both ILI and SARI cases)<br>3. Health facilities in the district report respiratory symptoms to the district level<br>• *Note: Response to any two of the questions makes it representative* |
| **Data quality** | - Completeness and accuracy of about 30 randomly selected case-based form, data from the sentinel site, DHIMS 2 and NIC data | Yes | $\%\ blankness = \frac{Number\ of\ blank\ spaces}{Total\ fields\ in\ dataset\ or\ based\ forms} \times 100$<br>* 71%–100% = Good data quality<br>* 51%–70% = Moderate quality<br>* 0%–50% = Poor quality |
| **Acceptability** | - Completeness of case-forms by staff<br>- Consistent collection of ILI sample or data within the past 3–6 months | Yes<br>Yes | - Based on the results/ grading of system's data quality;<br>* 71%–100% = Acceptable<br>* 51%–70% = Moderately acceptable<br>* 0%–50% = Not Acceptable<br>- Consistent data collection for 3–6 months = Acceptable<br>*Note: Yes to both questions make the system acceptable* |
| **Flexibility** | - Accommodates other events (e.g., COVID-19)<br>- Skilled staff easily adapts to changes in case definition | Yes<br>Yes | - Flexible = if system;<br>1. Accommodates other events or other respiratory events has been integrated (through the review of the newly improved ILI protocol and data from NIC and the sentinel site)<br>2. Staff adapts to changes in case definition<br>• *Note: Yes to both questions makes the system flexible*<br>• *Yes to only one question makes the system moderately flexible* |
| **Stability** | - Involvement of local, national and international agencies<br>- Availability of regular funding and logistics, reagents for laboratory testing<br>- Allocation of funds by Government (MoH) | Yes<br>Yes<br>Yes | Stable = Yes to either of the three questions grades the system as stable<br>• Note: Moderately stable: overdependency on international agency |
| **Sensitivity** | - System detected any epidemics in the past 4years<br>- System able to pick all cases | Yes<br>Yes | Sensitive = Yes to either of the two questions |
| **Timeliness** | Average time between:<br>- sample collection and submission for laboratory test<br>- feedback of results to all surveillance levels | Yes<br>Yes | - Timely = System;<br>1. Collects sample and submit to laboratory (NIC) within 48 hours' time approved in the ILI protocol<br>2. Reports or sends feedback to all surveillance levels within 48 hours' time approved in the ILI protocol<br>• *Note: System is timely if yes to both questions*<br>• *Not timely, if yes to only questions* |

## Description of the ILI surveillance system in Okai Koi North District

Through our stakeholder interviews, we found how the ILI surveillance system operates in the district, which has one influenza sentinel site (facility). The ILI surveillance system in Okai Koi North District operates specifically from the health facilities, particularly the Flu sentinel site, where all patients presenting with respiratory infections are approached and triaged. The sentinel site collects ILI samples from infected patients and sends to the NIC to be tested and confirmed for influenza. The non-sentinel sites (health facilities in the district) on the other hand do not collect samples, they only report upper respiratory tract infection (URTI), which also presents symptoms resembling influenza-like-illness, to the district level to be entered into DHIMS 2. The non-sentinel sites only collect respiratory samples on rare occasions when there are unexpected case reports and sends to the DHD to be transported to the NIC for confirmation.

At the Flu sentinel site, clinicians suspect or clinically diagnose patients based on the ILI case definition and the health information officers and disease control officers are alerted to follow up these patients for epidemiological data collection into the DHIMS2 or notifications of unusual occurrences to the next level, under the Ghana Health Service, thus the Disease Surveillance Department (DSD-GHS). They also collect ILI sample and transport within 48 hours on ice, to the NIC for laboratory testing and confirmation. First five ILI cases that presents at the sentinel site and meets the WHO approved case definition for ILI are expected to be sampled weekly. From in-patient department, all suspected cases samples are expected to be sampled for testing. The NIC sends Research Assistants out twice weekly (Tuesdays and Thursdays) to retrieve samples accompanied by filled-out case-based forms. Swab samples received by the NIC are processed/tested for the presence of influenza virus by polymerase chain reaction (PCR) and virus isolation. Results feedback are sent on weekly basis from the NIC to the sentinel site and DHD and in rare occasions, if other non-sentinel sites report, results are sent through the sentinel site. The NIC also sends results weekly into GHS and to WHO platforms, FluNet. The summary of the system operation and the flow of information and feedback is summarized in Fig 2.

**Case definition.** Influenza-like Illness (ILI)

- Any person with sudden onset of fever ≥ 38°C (axillary) and cough with onset within the last 10 days

## Performance of the system

**Usefulness.** Findings from the ILI sentinel site data review showed an all-year case detection, mostly in a timely manner and shown a trend of disease although there was no established threshold used by the site nor district for epidemic preparedness. The system generated influenza data which characterize, subtype, and describe circulating virus in the district. Majority of the respondents, 92.3% (12/13) affirmed this by their response to the morbidity question. They indicated that the ILI system in the district estimates morbidity however, the data does not show any estimates of mortality. The circulating influenza pathogen in the district included Flu A, 74% (43/58) and Flu B, 26% (15/58) with specifics to A(H1N1), A(H3N2), Flu B Victoria and Flu B Yamagata (Table 2). The system detected epidemics over the 4-year period, evidenced from the C-2 threshold developed using the sentinel site data (Fig 3). Similarly in 2021, during the evaluation period, the district detected SARS-CoV-2 virus, during the heat of the COVID-19 pandemic.

Although the district nor the sentinel site had no developed threshold for ILI epidemics, we developed a C-2 threshold for the 4 -year evaluation period. It was found that in December 2019, April 2021 and July 2021, the system was able to detect influenza epidemics (Fig 3). Even in 2021 when the system integrated COVID-19 into the ILI surveillance, the system was still able to pick up epidemics.

The influenza pathogens detected over the 4-year period at the sentinel site are shown in Table 2 below.

## Simplicity

Majority, 92% (12/13) of staff interviewed indicated that, the case definition for surveillance is simple. A total of 77% (10/13) of them also revealed that case-based form is easy to fill. However, it was found that the system required

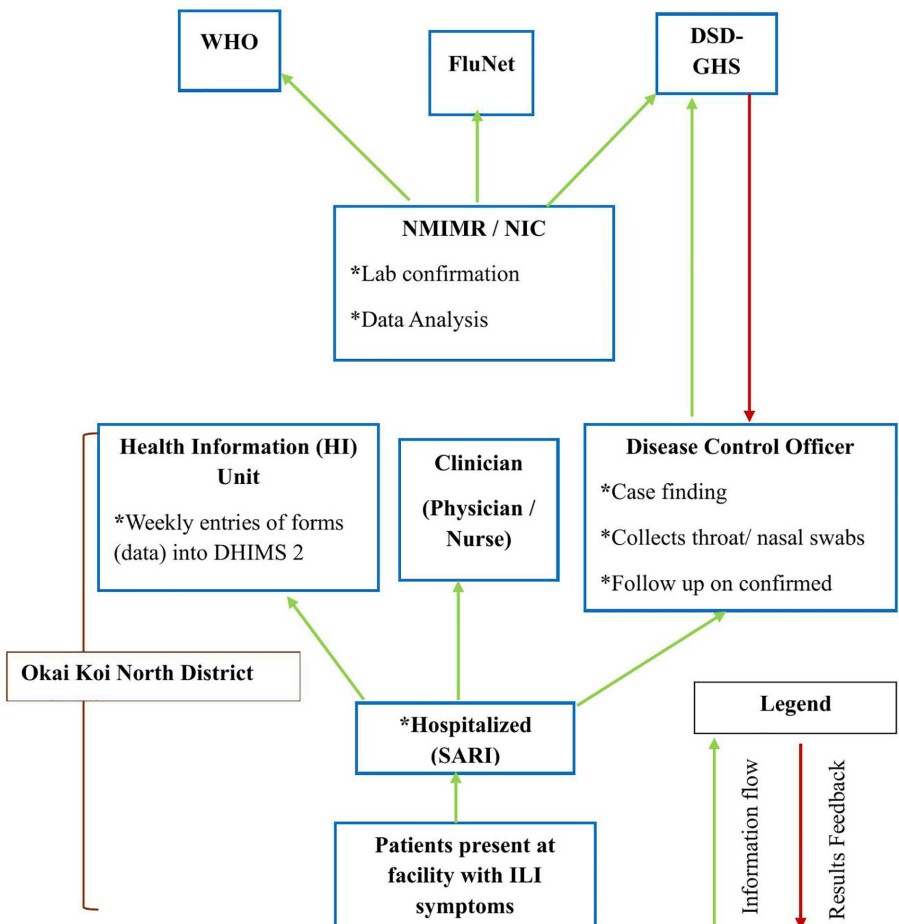

**Fig 2. Flow chart of information and feeback flow within the ILI surveillance system, Okai Koi North District, 2022.**

**Table 2. Influenza detection at the Flu sentinel site, Achimota Hospital (Okai Koi North District), 2018-2021.**

| Year | Number of suspected ILI cases | Flu positives n (%) | Characterization of Flu Positives | | | | | |
|---|---|---|---|---|---|---|---|---|
| | | | Subtypes | | | Lineages | | |
| | | | Flu A | A(H1N1)pdm09 | A(H3N2) | Flu B | Yama-gata | Victoria |
| 2018 | 207 | 29 (14) | 17 | 13 | 4 | 12 | 10 | 2 |
| 2019 | 48 | 5 (10) | 4 | 0 | 4 | 1 | 0 | 1 |
| 2020 | 147 | 2 (1) | 0 | 0 | 0 | 2 | 0 | 2 |
| 2021 | 123 | 22 (17) | 22 | 19 | 3 | 0 | 0 | 0 |
| **Total** | 525 | 58 (11) | 43 (74%) | 32 | 11 | 15 (26%) | 10 | 5 |

laboratory for testing, specifically using PCR and viral isolation for confirmation and characterization. Samples collected at the sentinel site are sent to NIC to be processed and to be confirmed for the existence of influenza pathogen.

## Positive predictive positive

For the 4-year period, there was an overall 58 influenza cases confirmed by the NIC from the sentinel site out of the 525 suspected cases. This gave a positive predictive positive value (PVP) of 11%.

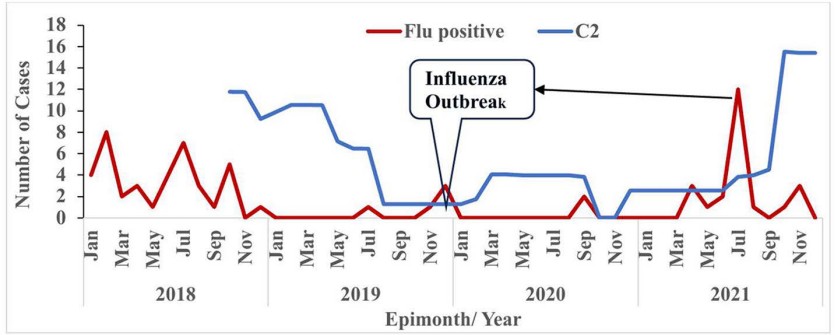

**Fig 3. Threshold showing the distribution of epidemics detected in Okai Koi North District, at the sentinel site, 2018-2021.**

## Representativeness

Data from the sentinel site reported cases from both males and females across all age-groups, however, distribution of the cases were mainly among young adults from 15–34 years as shown in Table 3. Data from the DHD and the sentinel site indicated ILI samples are collected from suburbs and surrounding environs in the district. For the five non-sentinel sites visited, data from their OPD records, DHIMS 2 and other registers revealed that URTI epidemiological data (just the number of URTI cases seen at the facility) are collected also from both males and females across all age-groups and are reported directly into the DHIMS 2.

## Data quality

We found 101 available filled-out case-based forms from the sentinel site of which 30 (10%) were randomly selected. There were 29 fields to be completed on each case-based form. Of the total 870 fields (29 fields by 30 case-based form) to be completed, there were 11% (99/870) blanks; thus representing 89% (771/870) completeness.

Data obtained from the sentinel site also showed 89% data completeness. Some discrepancies were noted in the data garnered from the sentinel site, district, regional and the national level. Data from the sentinel site was harmonized with the data from the NIC (from the district) however, there were differences recorded in the year 2021 in the evaluation period. The aggregated data from the sentinel site sampled 201 ILI cases in the year 2021, however, samples received, tested, and entered in the NIC database at the national level were 123 (61%) cases. At the non-sentinel sites, there were

**Table 3. Age distribution of population under ILI sentinel surveillance, Influenza sentinel site, Okai Koi North District, Accra, Ghana, 2018–2021.**

| Age group | Number of suspected cases | Number of confirmed positives | Number of Negatives |
|---|---|---|---|
| Total | 525 | 58 | 467 |
| 0-4 | 7 | 2 | 5 |
| 5-14 | 35 | 9 | 26 |
| 15-24 | 128 | 16 | 112 |
| 25-34 | 120 | 16 | 104 |
| 35-44 | 84 | 8 | 76 |
| 45-54 | 39 | 2 | 37 |
| 55-64 | 39 | 0 | 39 |
| ≥65 | 46 | 3 | 43 |
| Unknown | 27 | 2 | 25 |

111,224 URTI cases reported at the district, however, only 13,842 were reported at the regional level. There were differences in the ILI recorded data from the district in DHIMS 2 and from NIC. Data from DHIMS 2 reported 178, 2, 0, 14 suspected ILI cases within 2018–2021 respectively. However, data from the sentinel site reported by the NIC were 207, 48, 147 and 123 respectively for 2018–2021. None of the completed case-based forms at the sentinel site, DHD and the NIC were updated with the results from the laboratory test. They also did not have results for final patients' outcome and classification.

## Acceptability

Data from the sentinel site revealed an 89% completeness of the case-based forms within the 4-year evaluation period. Also, from the sentinel site data, it was found that there has been a diligent participation of surveillance staff as seen in the consistent, all year-round collection of Flu-like samples, for 4-year period evaluation period. This was affirmed by majority of respondents, 85% (11/13).

## Flexibility

All respondents indicated the surveillance system adapted very easily when new symptoms are updated to the surveillance case definition for ILI, in the event of new health demands. This was evident as the system used the ILI case definition at the initial stages of the COVID-19 pandemic. Currently, Ghana has adopted and integrated SARS-CoV 2 and other non-influenza-like viruses [e.g., respiratory syncytial virus (RSV)] into its ILI surveillance system. Again, all the surveillance staff interviewed indicated skilled staff adapted easily to the changes in case definition during the early phases of the COVID-19 pandemic. Only 31% (4/13) of the staff iterated that variation in focal person usually affects the ILI surveillance system's performance.

## Stability

A total of 85% (11/13) of the surveillance staff interviewed indicated there is regular funding for the system operation. The ILI surveillance system at the Okai Koi North District has the involvement of local, national, and multinational agencies in the surveillance system through logistics, salaries, and capacity building. These agencies include GHS, the Government of Ghana, WHO, and United States centers for disease control and prevention (US CDC). Interview with the NIC focal person indicated a constant reagents and logistics provision by WHO, US CDC, NAMRU-3, and other external collaborators, for the operation of surveillance activities. It was found that aside from the payment of health facility and district staff salaries, the ILI system does not have government allocated funds or budgets.

## Sensitivity

Although the ILI surveillance system has no established thresholds levels, data from the sentinel site found that the system was able to detect ILI cases and outbreaks in the 4-year evaluation period (Fig 4).

Although, there was a halt in the collection of ILI cases at the sentinel site during the initial phase of the COVID-19 pandemic (from April – June 2020), however, the system incorporated COVID-19 case sampling. In 2021, the system detected ILI cases amidst COVID-19.

## Timeliness

From the ILI sentinel data from the district, the average time between symptoms onset in patients and reporting to the site (health facility) for treatment was 6 + 7 days. Furthermore, an average of 3 ± 4 days (48 hours) elapsed from time samples were collected at the sentinel site, to the time they were received at the laboratory (NIC) for confirmatory testing. A total of 306/525 (58%) of the samples were transported within 48 hours meeting the set target days. Key informant interview

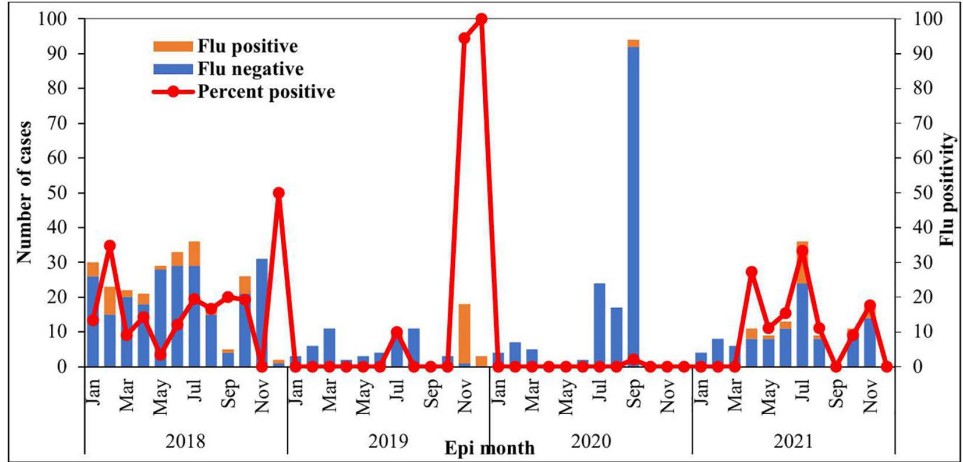

**Fig 4. Distribution of Influenza cases with its positivity from the sentinel site, Okai Koi North District, 2022.**

from NIC revealed that it took an average of two days to process and test samples when reagents are available. Results of samples tested were sent by email to the sentinel site on weekly basis. For the non-sentinel sites, it was found that staff reports aggregated data on URTI to the district level (DHD), via text messages and phone calls. About 83% (5/6) of the respondents indicated reception of results feedback from NMIMR.

## Discussion

We highlight findings from the evaluation of the ILI surveillance system of Okai Koi North district for the performance of the system, in relation to its usefulness and the system attributes. The ILI surveillance in the Okai Koi North District generally is useful as most of its surveillance objectives are met. The system with the full participation of laboratory system, in the use of PCR is sensitive enough in its ability to detect influenza outbreaks. This finding corroborates with findings by Ameme et al., (2020) who reported influenza outbreaks in the main senior high school (SHS) in Okai Koi North District. In the broader light, the ILI surveillance operated in Ghana has helped detect outbreaks across the nation, as reported by researchers [9,13]. The system is able to characterize the circulating strains among the population in the district. This result agrees with Nuvey et al., (2019) who reported that influenza virus types A and B were the most common kinds of circulating influenza in Ghana when he evaluated three influenza sentinel sites. This largely depicts a global image, which has a comparable distributional trend.

Threshold development serves as an early warning system for Flu-like outbreaks in the country [14]. However, the evaluation saw no threshold developed for the ILI surveillance system in the district. The nationwide ILI surveillance system has no thresholds even though it was recommended by Nuvey et al., (2019) in a similar study. Setting thresholds for disease surveillance is essential to enabling the health systems to quickly take public health action when outbreaks happen. A national implementation of threshold development for the ILI surveillance system could be achieved through stakeholder engagement, to review the national influenza protocol for the inclusion of a mandatory development of thresholds, at national, regional, district and sub-district levels. Hence include national level training of all surveillance staff at the lower levels on thresholds development, data entries for its development and interpretations of thresholds. Also, SORMAS (surveillance, Outbreak Response Management and Analysis System), a data platform used by GHS could have its dashboard updated for the calculation of thresholds.

The system estimates morbidity burden however the surveillance data does not show any mortality estimates. This finding is different from Chaves et al., (2013) as it reports on influenza severity and mortality [15]. The ILI surveillance system could improve or integrate data entry into SORMAS at the facility level, which captures the epidemiological data of the patient and thus, be linked to the data entry platform at the NIC for monitoring and subsequent completion of all the required fields after the results are confirmed. Both sides complete all required fields including disease outcomes which foster proper monitoring. This plan implemented by the stakeholders at the national level will help understand disease severity such as mortality, recovery, lost to follow-up, etc.

Seasons have significant differences in the influenza burden of illness. The Okai Koi North District in Ghana also has a relatively undefined transmission pattern as reported by Radin and colleagues in their work [16], thus suggestive of the Flu seasonality in the country. Influenza seasonality is probably influenced by several variables, such as climate, host distribution, and behavioral patterns. The transmission of Influenza is largely influenced by chilly temperatures, low relative and absolute humidity, according to laboratory studies by Radin and colleagues.

Uncertainty surrounds how these factors contribute to Ghana's lack of influenza seasonality. It only demonstrates the necessity for additional research conducted over a longer time span to look for a relationship between the flu virus and seasonality in terms of the local climate, as it was suggested by an influenza resource person in a WHO bulletin [17]. The impact of seasonal influenza in the local environment will be brought to light by estimates of the national disease burden during influenza seasons, starting from the sub-district levels. It may encourage greater efforts to contain the influenza outbreak in areas where the effects are most severe. Also researching the burden of disease associated with seasonal influenza also aids nations in building their analytical and surveillance capacities for use in pandemic situations as recommended by literature [1]. Notwithstanding the seasonal pattern of ILI circulation, we observed that, the system does not have influenza disease burden or mortality estimates which could help in providing in-depth understanding into the socioeconomic, clinical, political impacts as well as strengthen preparedness [18].

The surveillance system was a complex one mainly because of the involvement of the NIC laboratory in testing ILI samples, using PCR. The detection of a positive influenza virus follows up with subtyping to determine the respective influenza strain [influenza A(H1N1)pdm09, A(H3N2), Influenza B Victoria and Yamagata]. Budgell and colleagues reported in their study that some basic laboratory testing done before sending to national labs [19]. This was not so in our findings, as samples collected at the sentinel sites are transported directly to the NIC for confirmatory testing.

The low PVP of the ILI surveillance system in the Okai Koi North District is consistent with the findings from Ghana, and Madagascar [20,21]. Reasons that could account for this lower PVP can be attributed to the sensitivity level of the case definition – could pick other upper respiratory tract infections and other medical conditions with similar symptoms. Also, it takes an average of six days for an ILI patient to report to the health facilities upon an onset of illness. The influenza virus may have been degenerated as the days exceed the incubation period of 1–4 days. This could account for the low PVP for Okai Koi North District.

The ILI surveillance system in Okai Koi North District reported cases from both males and females, and persons across all age-groups. However, the case-based forms and the data does not indicate the specific at-risks groups (pregnancy, visit to endemic places) which is contrary to findings by [22–25] which reported population at risk of ILIs in their studies. Bonney et.al confirmed severe form of ILI infection among children under 11 years [24] in Ghana however, in our evaluation, influenza virus infection was most prevalent among persons ages 15–24 years which could be due to the boarding SHS in the district; consistent with Ameme *et, al.*, (2020) report.

The reported data discrepancy between the sentinel site and the NIC in our study could be due to non-existing routine/periodic data harmonization, samples received at the NIC without case investigation forms, and insufficient sample volume due to leakage. This discrepancy could also be as a result of the surge and prioritization for the testing of COVID-19 cases in 2021, where some ILI cases from the district could have been treated as COVID-19 cases. Efforts aimed at curbing data discrepancies in the ILI system could be through an implemented and routine stakeholder quarterly meeting,

between both national (GHS and NIC) and district (sentinel site). These routine meetings could be held at the sentinel site where missing information including patients' outcome, are reconciled, using the source data (i.e., patients' folders). It also offers an opportunity for the facility to update their forms or data routinely with the results from the NIC lab. More robustly, the adaptation of SORMAS at the national level which was used during the COVID-19 era could be tweaked to include the capture of influenza data entry interface at all levels as well as at the NIC. The adaptation of SORMAS will foster the mandatory/required completion of all essential data variable on the questionnaire [26]. Through SORMAS, the NIC would be able to update the specimen test results field in real-time for use at the district or health facility level to aid patient care. To resolve the issue of insufficient sample volumes the NIC and stakeholders at the national level should periodically train lower-level staff and research assistants that transport samples, on specimen handling. As the NIC continues to supply the Flu site with VTM (viral transport medium), they need to train both their research assistants and the staffs at the sentinel site on the proper cocking of VTMs and thus, safe handling during sample collection and transportation. Whereas COVID-19 samples were prioritized during the pandemic, we will recommend that during the event of outbreaks and pandemics, laboratory testing which are associated with pathogens of the same family could be conducted concurrently.

This evaluation found surveillance staff acceptance and willingness to participate in surveillance activities as they consistently collected ILI samples for all-year round with 89% data completeness. The staff had good understanding of the ILI protocols, SOPs and case-definition. These could account for their acceptance and willingness to work during surveillance activities as well as the high data completeness. These findings are similar to researchers' findings [20,21].

Furthermore, the ILI system in Okai Koi North District, had a major strength in flexibility. During the inception COVID-19 pandemic, the system with skilled staff was able to quickly adapt and accommodate changes made in the previous ILI case definition, which integrated the detection of SARS-CoV 2 virus and other respiratory pathogens including syncytial virus (RSV). This can be ascribed to the simplicity and flexibility of the case definition, corroborated with findings in previous evaluations in Madagascar and South Africa [19,21].

The system involves both local, national, and international stakeholders with over-dependence or operations mostly funded by international donors. It was found that apart from the salaries paid to the district and the health facility staff, there was no budget or funds made available for the ILI surveillance system. The overdependence of external or international donors may reveal gaps in the surveillance system that must be addressed. This was evident in Nuvey et. al., (2019) report where there was an interruption in the NIC workflow in the laboratory testing during shortage of reagents. In as much as the NIC and the influenza surveillance system in Ghana, has over the years depended on funds from external donors, such as the US CDC, WHO and NAMRU-3 long-terms sustainability of the system remains grey. To resolve the overdependency on external donors, sustainability efforts could be through advocacy to highlight the importance of the ILI surveillance system to the government, for the consideration of budget allocation to support equipment procurement/ service contracts, reagents and consumables all year round. During the COVID-19 era, the NIC experienced a novel and heightened private-public partnership including financial institutions, private institutions contributing towards the procurement of equipment, donation of reagents and consumable [27–33]. The ILI surveillance system could leverage on this cemented private-public partnership, to advocate for financial support as well as grants towards equipment procurement and service contracts, reagents, and consumables for infection-prevention-control (IPC).

Findings from literature by Rakotoarisoa et al., (2017) on timeliness is contrary to what was found in this evaluation, where a higher proportion (58%) of samples collected was sent to the lab within 48 hours. Reason that could account for the timely collection of samples and reception at the laboratory for confirmatory testing, might be the visit by NIC research to the sentinel sites within the Greater Accra region for specimen retrieval and the proximity of the sentinel site (Achimota Hospital) and the NIC unlike courier services which take longer hours and days as seen in Rakotoarisoa report. Although our findings indicate an average of 48 hours for specimen testing, weekly submission of results and reports to sentinel sites and global platform, system does not capture data on date of specimen testing at the NIC which refutes the calculation of turn-around-time of specimen testing which could impact result feedback for patient care [34,35]. Data

reconciliation as proposed earlier using the SORMAS platform could remedy this gap, where specific date of testing of patients' specimen could be completed.

Findings from our study indicated that the NIC shares weekly reports to the global FluNet platform, however, epidemiological data from all ILI suspected cases (sampled and not sampled) are not shared into FluID (WHO platform for ILI epidemiological data). The ILI surveillance systems inability to submit data into FluID to complement the FluNet data unfavorably affect the tracking of global trends, spread, intensity, disease burden estimates and impact of influenza [36].

Due to time constraints, other influenza sentinel sites in the Greater Accra Region could not be included in the evaluation, hence we could not compare the ILI activities and findings from Okai Koi North District to other sentinel site activities and findings. Despite the focus on Okai Koi North District, findings from this study could largely help improve existing ILI surveillance systems since most of the districts in Ghana has similar characteristics especially within the urban regions. The findings from this study could also serve as a baseline data to structure new ILI surveillance systems in other districts of Ghana and sub-Saharan Africa.

Globally, disease surveillance is taking a new turn with new approaches and technologies such as participatory surveillance and web-based surveillance which could improve participation in the traditional ILI surveillance in Ghana. Participatory surveillance (PS) is an evolving strategy which when incorporated, actively engages community members and support community volunteers to improve reporting on health incidents on a regular basis to obtain data to monitor trends. In the broader perspective, PS may reveal indications of an early outbreak by showing signs of an emerging disease or a concentration of symptoms in particular locations especially hard to reach communities. On the other hand, web-based surveillance is another direction of public health surveillance system that leverages on web-applications platform such as, social media, twitter, to track disease patterns with the Ghanaian population using aggregate data from search queries [37,38].

## Conclusion

Overall, the ILI surveillance system in Okai Koi North District was useful as most of its objectives were met and pivotal in the diagnosis and response to the COVID-19 pandemic in Ghana. The system is sensitive to detect epidemics and identify circulating strains of influenza in the district, flexible as it adapted the integration of COVID-19 and other pathogens detection. It is timely, representative of the population under study, and has significant data quality. However, the system is complex as it requires laboratory for confirmation and has low PVP.

In spite of the sensitivity of the system, there are no thresholds set for the system to run, hence key stakeholders at the national level, thus from the Disease Surveillance Department-GHS (DSD-GHS) and NIC should collaborate and develop national influenza threshold serves as an early warning system for influenza outbreaks in the country. This will further prevent missing out on epidemics at sentinel and non-sentinel sites. At the district and health facility levels health information (HI) unit responsible for data analysis and management could be rigorously trained by the data managers at the NIC or national level on the use of Microsoft Excel to generate and chart the ILI and other diseases' threshold alerts. Epidemiological threshold levels at the healthy facility levels should be generated by epidemiological weeks to support the early detection of surge in ILI cases, which will prompt heightened surveillance through active case search. With enhanced text messaging features incorporated in Ghana health service's data platforms, DHIMS-2 and SORMAS, the HI teams could receive and share prompt messages when an alert/epidemic threshold is met.

Though the system has good data quality, the incompleteness of important fields on the case-based forms reveals a system gap. Hence, surveillance staff and NIC staff should mandate laboratory results, final outcome of patients and classification as important fields to be filled.

## Supporting information

**S1 Fig. Map of health facilities and institutions visited for the ILI surveillance system evaluation, Okai Koi North District, 2022.**
(TIF)

**S2 Fig. Flow chart of information and feedback flow within the ILI surveillance system, Okai Koi North District, 2022.**
(TIF)

**S3 Fig. Threshold showing the distribution of epidemics detected in Okai Koi North District, at the sentinel site, 2018–2021.**
(TIF)

**S4 Fig. Distribution of Influenza cases with its positivity from the sentinel site, Okai Koi North District, 2022.**
(TIF)

**S5 File. Questionnaire and checklist for the assessment of the ILI surveillance system.**
(DOCX)

## Acknowledgments

Our sincere gratitude and appreciation go to the Okai Koi North District Health Directorate Team of the Ghana Health Service; Nana Ama Adjabeng, Joyce Boatemaa Yeboah, Isaac Buabeng Bempong and Felicia Brako, for giving us access to use their data sets. Another appreciation goes to the data team at the National Influenza Center of the Noguchi Memorial Institute for Medical Research for enabling us to use the relevant district's data sets. Also, we appreciate all the surveillance staff at the various health facilities visited and both staff and mentors of GFELTP.

## Author contributions

**Conceptualization:** Doris Aboagyewaa Edu-Quansah, Delia Akosua Bandoh, Elijah Paa Edu-Quansah, Charles Lwanga Noora, Ivy Asantewaa Asante, Dennis Laryea, Ernest Kenu.

**Data curation:** Elijah Paa Edu-Quansah.

**Formal analysis:** Doris Aboagyewaa Edu-Quansah, Delia Akosua Bandoh, Elijah Paa Edu-Quansah.

**Investigation:** Doris Aboagyewaa Edu-Quansah.

**Methodology:** Doris Aboagyewaa Edu-Quansah, Delia Akosua Bandoh, Anthony Baffour Appiah, Ivy Asantewaa Asante.

**Resources:** Anthony Baffour Appiah, Charles Lwanga Noora, Ivy Asantewaa Asante, Dennis Laryea, Ernest Kenu.

**Supervision:** Delia Akosua Bandoh, Elijah Paa Edu-Quansah, Anthony Baffour Appiah, Charles Lwanga Noora, Ivy Asantewaa Asante, Dennis Laryea, Ernest Kenu.

**Writing – original draft:** Doris Aboagyewaa Edu-Quansah, Elijah Paa Edu-Quansah, Ernest Kenu.

**Writing – review & editing:** Delia Akosua Bandoh, Elijah Paa Edu-Quansah, Anthony Baffour Appiah, Charles Lwanga Noora, Ivy Asantewaa Asante, Dennis Laryea, Ernest Kenu.

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
