## [Decision Letter · Decision Letter 0]

25 Mar 2025

Dear Dr. Edu-Quansah,

**Please take into account all remarks made the Referee, especially regarding the clarity of the presentation.**

We look forward to receiving your revised manuscript.

Kind regards,

Michele Tizzoni

Academic Editor

PLOS ONE

**Journal Requirements:**

1. When submitting your revision, we need you to address these additional requirements. Please ensure that your manuscript meets PLOS ONE's style requirements, including those for file naming. The PLOS ONE style templates can be found at https://journals.plos.org/plosone/s/file?id=wjVg/PLOSOne_formatting_sample_main_body.pdf and https://journals.plos.org/plosone/s/file?id=ba62/PLOSOne_formatting_sample_title_authors_affiliations.pdf 2. You indicated that ethical approval was not necessary for your study. We understand that the framework for ethical oversight requirements for studies of this type may differ depending on the setting and we would appreciate some further clarification regarding your research. Could you please provide further details on why your study is exempt from the need for approval and confirmation from your institutional review board or research ethics committee (e.g., in the form of a letter or email correspondence) that ethics review was not necessary for this study? Please include a copy of the correspondence as an "Other" file.  3. We note that your Data Availability Statement is currently as follows: All relevant data are within the manuscript and its Supporting Information files. Please confirm at this time whether or not your submission contains all raw data required to replicate the results of your study. Authors must share the “minimal data set” for their submission. PLOS defines the minimal data set to consist of the data required to replicate all study findings reported in the article, as well as related metadata and methods (https://journals.plos.org/plosone/s/data-availability#loc-minimal-data-set-definition). For example, authors should submit the following data: - The values behind the means, standard deviations and other measures reported;- The values used to build graphs;- The points extracted from images for analysis. Authors do not need to submit their entire data set if only a portion of the data was used in the reported study. If your submission does not contain these data, please either upload them as Supporting Information files or deposit them to a stable, public repository and provide us with the relevant URLs, DOIs, or accession numbers. For a list of recommended repositories, please see https://journals.plos.org/plosone/s/recommended-repositories. If there are ethical or legal restrictions on sharing a de-identified data set, please explain them in detail (e.g., data contain potentially sensitive information, data are owned by a third-party organization, etc.) and who has imposed them (e.g., an ethics committee). Please also provide contact information for a data access committee, ethics committee, or other institutional body to which data requests may be sent. If data are owned by a third party, please indicate how others may request data access. 4. We note that Figure 1 in your submission contain map images which may be copyrighted. All PLOS content is published under the Creative Commons Attribution License (CC BY 4.0), which means that the manuscript, images, and Supporting Information files will be freely available online, and any third party is permitted to access, download, copy, distribute, and use these materials in any way, even commercially, with proper attribution. For these reasons, we cannot publish previously copyrighted maps or satellite images created using proprietary data, such as Google software (Google Maps, Street View, and Earth). For more information, see our copyright guidelines: http://journals.plos.org/plosone/s/licenses-and-copyright. We require you to either present written permission from the copyright holder to publish these figures specifically under the CC BY 4.0 license, or remove the figures from your submission: a. You may seek permission from the original copyright holder of Figure 1 to publish the content specifically under the CC BY 4.0 license.   We recommend that you contact the original copyright holder with the Content Permission Form (http://journals.plos.org/plosone/s/file?id=7c09/content-permission-form.pdf) and the following text:“I request permission for the open-access journal PLOS ONE to publish XXX under the Creative Commons Attribution License (CCAL) CC BY 4.0 (http://creativecommons.org/licenses/by/4.0/). Please be aware that this license allows unrestricted use and distribution, even commercially, by third parties. Please reply and provide explicit written permission to publish XXX under a CC BY license and complete the attached form.” Please upload the completed Content Permission Form or other proof of granted permissions as an "Other" file with your submission. In the figure caption of the copyrighted figure, please include the following text: “Reprinted from [ref] under a CC BY license, with permission from [name of publisher], original copyright [original copyright year].” b. If you are unable to obtain permission from the original copyright holder to publish these figures under the CC BY 4.0 license or if the copyright holder’s requirements are incompatible with the CC BY 4.0 license, please either i) remove the figure or ii) supply a replacement figure that complies with the CC BY 4.0 license. Please check copyright information on all replacement figures and update the figure caption with source information. If applicable, please specify in the figure caption text when a figure is similar but not identical to the original image and is therefore for illustrative purposes only.The following resources for replacing copyrighted map figures may be helpful: USGS National Map Viewer (public domain): http://viewer.nationalmap.gov/viewer/The Gateway to Astronaut Photography of Earth (public domain): http://eol.jsc.nasa.gov/sseop/clickmap/Maps at the CIA (public domain): https://www.cia.gov/library/publications/the-world-factbook/index.html and https://www.cia.gov/library/publications/cia-maps-publications/index.htmlNASA Earth Observatory (public domain): http://earthobservatory.nasa.gov/Landsat:
http://landsat.visibleearth.nasa.gov/USGS EROS (Earth Resources Observatory and Science (EROS) Center) (public domain): http://eros.usgs.gov/#Natural Earth (public domain): http://www.naturalearthdata.com/

Reviewers' comments:

Reviewer's Responses to Questions

**Comments to the Author**

1. Is the manuscript technically sound, and do the data support the conclusions?

Reviewer #1: Partly

2. Has the statistical analysis been performed appropriately and rigorously?

Reviewer #1: No

3. Have the authors made all data underlying the findings in their manuscript fully available?

Reviewer #1: No

4. Is the manuscript presented in an intelligible fashion and written in standard English?

Reviewer #1: Yes

**Reviewer #1:**  The paper “Evaluation of the Performance of the Influenza-like Illness (ILI) Surveillance System in the Okai Koi North District, Greater Accra Region, 2022,” aims to assess how well the ILI surveillance system meets its stated goals and CDC/WHO-recommended attributes (e.g., timeliness, data quality, representativeness, etc.). It also attempts to gauge the system’s usefulness in detecting and characterizing influenza in the district over a four-year period (2018–2021). Overall, the manuscript is well structured and follows the CDC’s updated guidelines for evaluating public health surveillance systems. It shows a balanced mix of quantitative and qualitative data and provides a clear picture of how the surveillance system operates, including stakeholder roles, laboratory processes, and data flow. However, it shows some gaps that should be addressed before being in a shape for acceptance.

Major review points:

1. In the introduction, provide a crisper statement of the study’s main objectives in the final paragraph to guide readers toward the methods. And consider emphasizing the importance of thresholds in the context of outbreak detection slightly earlier, so that readers understand the significance of that gap when it appears later in the paper.

2. Strengthen the explanation of how thresholds were retroactively generated using CUSUM.

3. Provide further explanation on the mismatch between sentinel site numbers and NIC records, including potential reasons (e.g., lost samples, incomplete form submission).

4. Provide a short guidance or example for how local health facilities could routinely apply thresholds (C-2, CUSUM, etc.) to perform outbreak detection, since a minimum background of disease surveillance and data analysis would be required for that, what implies in granting local capacity and workforce towards this training. Also, how early warning system approach could be incorporated in this.

5. Addressing in a more detailed way despite the limited generalizability due to single-district focus, how these findings could be instrumental for improving ILI surveillance in similar settings.

6. Since potential recall or reporting biases are possible in self-reported data about timeliness and system operations were possible, what be the directions to overcome this during building author's conclusions?

7. An addition of future perspectives for ILI surveillance could be helpful for readers understanding towards the evolving process of this area. The following reference is just one of examples on how ILI surveillance have been developed in new aspects: https://publichealth.jmir.org/2023/1/e46644/

Minor review points:

8. Watch for typographical inconsistencies (e.g., “Okai Koi North” vs. “Okaikoi North”

9. Ensure all acronyms (GISRS, NIC, DHD, etc.) are spelled out at first mention.

**Do you want your identity to be public for this peer review?** For information about this choice, including consent withdrawal, please see our Privacy Policy

Reviewer #1: **Yes: ** Onicio B. Leal Neto

---

## [Author Response · Author response to Decision Letter 1]

15 May 2025

Rebuttal letter addressing each point raised by academic editors and reviewers

Major review points:

1. In the introduction, provide a crisper statement of the study’s main objectives in the final paragraph to guide readers toward the methods. And consider emphasizing the importance of thresholds in the context of outbreak detection slightly earlier, so that readers understand the significance of that gap when it appears later in the paper.

Response: The study’s main objective has been made more crisper in the final paragraph of the introduction, found on lines 66-68 of the introduction. It reads as “We present here findings on our evaluation of the ILI surveillance system in Okai Koi North District in assessing the systems performance in concert with its usefulness and system attributes”.

The importance of threshold has also been included earlier in the introduction, in the context of outbreak detection. This can also be found on line 49-56 of the introduction and it reads as, “These Flu outbreaks usually meet the lower levels of surveillance, example the district levels unprepared to mitigate them which either goes undetected or delayed detection. The Flu situation is not the same for all diseases, as some diseases under surveillance here in Ghana, example Measles have epidemic thresholds set for them. Thresholds denote a degree of incidence that is essential for timely detection of disease outbreaks, and its response”.

2. Strengthen the explanation of how thresholds were retroactively generated using CUSUM.

Response: The generation of thresholds for the district using CUSUM-C2 method has been explained into details at the data analysis section under materials and methods. It can be found on line 174-182 to read as follows;

“We generated an ILI epidemic threshold for Okai Koi District from data garnered from the NIC using the CUSUM - C2 method. The four years data were arranged by year and months (January – December). For each threshold point, we took the mean and standard deviation for a group of seven surveillance points (months over the four years) in the past, skipping over the two most recent points (months). We then added three times the standard deviation to the mean value to obtain the C2 value for specific point/month [C2 = Mean + 3(standard deviation)]. The calculated C2 values for the months were used to plot the epidemic threshold chart using Microsoft Excel”.

3. Provide further explanation on the mismatch between sentinel site numbers and NIC records, including potential reasons (e.g., lost samples, incomplete form submission).

Response: This has been further explained and reasons allocated for it under the discussion section, on line 524-528. And it reads as follows,

“The reported data discrepancy between the sentinel site and the NIC by our study could be due to non-existing routine/periodic data harmonization, samples received at the NIC without the case investigation forms, and insufficient sample volume due to leakage. This discrepancy could also be as a result of the surge and prioritization of COVID-19 cases in 2021, where some ILI cases from the district could have been treated as COVID-19 cases”.

4. Provide a short guidance or example for how local health facilities could routinely apply thresholds (C-2, CUSUM, etc.) to perform outbreak detection, since a minimum background of disease surveillance and data analysis would be required for that, what implies in granting local capacity and workforce towards this training. Also, how early warning system approach could be incorporated in this.

Response: This has been addressed under conclusion, line 601-620, to read as,

“At the district and health facility levels health information (HI) unit responsible for data analysis and management could be rigorously trained by the data managers at the NIC or national level on the use of Microsoft Excel to generate and chart the ILI and other diseases’ threshold alerts. Epidemiological threshold levels at the healthy facility levels should be generated by epidemiological weeks to support the early detection of surge in ILI cases, which will prompt heightened surveillance through active case search. Ghana health service has data platforms for every surveillance level, such as DHIMS-2 and SORMAS (surveillance, Outbreak Response Management and Analysis System), with enhanced text messaging features incorporated in them, the HI teams could receive and share prompt messages when an alert/epidemic threshold is met”.

5. Addressing in a more detailed way despite the limited generalizability due to single-district focus, how these findings could be instrumental for improving ILI surveillance in similar settings.

Response: This has been addressed under the discussion section, line 568-573. It then reads as “Despite the focus on Okai Koi North District, findings from this study could largely help improve existing ILI surveillance systems since most of the districts in Ghana has similar characteristics especially within the urban regions. The findings from this study could also serve as a baseline data to structure new ILI surveillance systems in other districts of Ghana and sub-Saharan Africa”.

6. Since potential recall or reporting biases are possible in self-reported data about timeliness and system operations were possible, what be the directions to overcome this during building author's conclusions?

Response: We were cognizant about potential recall or reporting bias during stakeholders or surveillance staff interview hence to minimize or reduce the effect of recall bias in our study, we undertook the activities stated below. This can be found under the evaluation approach and data collection technique under the materials and methods section, on line 151-158 and it reads as follows,

“Surveillance data from the NIC, which is data from the sentinel site and DHIMS 2 were reviewed and extracted using the developed checklist. Also, we reviewed consulting room registers and OPD records from all the health facilities visited within the evaluation period, using the developed checklist. The evaluation targeted information from records (data and forms) on all in-patients and out-patients, males and females reporting at these visited health facilities in the district as well as data on samples received at the NIC. Similarly, surveillance activities were also observed using the checklists. The purpose of the observation carried out related stakeholders’ answers to questionnaire to what was practiced on the field”.

7. An addition of future perspectives for ILI surveillance could be helpful for readers understanding towards the evolving process of this area. The following reference is just one of examples on how ILI surveillance have been developed in new aspects: https://publichealth.jmir.org/2023/1/e46644/

Response: This has been addressed under the discussion section, on line 574-589, to read as,

“Globally, disease surveillance is taking a new turn with new approaches and technologies such as participatory surveillance and web-based surveillance could improve participation in the traditional ILI surveillance in Ghana. Participatory surveillance (PS) is an evolving strategy which when incorporated, actively engages community members and support community volunteers to improve reporting on health incidents on a regular basis to obtain data to monitor trends. In the broader perspective, PS may reveal indications of an early outbreak by showing signs of an emerging disease or a concentration of symptoms in particular locations especially hard to reach communities. On the other hand, web-based surveillance is another direction of public health surveillance system that leverages on web-applications platform such as, social media, twitter, to track disease patterns with the Ghanaian population using aggregate data from search queries.”

Minor review points:

8. Watch for typographical inconsistencies (e.g., “Okai Koi North” vs. “Okaikoi North”)

Response: This has been resolved in manuscript. Okai Koi North District has been used to replace all typographical inconsistencies regarding its name.

9. Ensure all acronyms (GISRS, NIC, DHD, etc.) are spelled out at first mention.

Response: All acronyms in manuscript have all been spelt out at first mention.

---

## [Decision Letter · Decision Letter 1]

4 Jun 2025

Dear Dr. Edu-Quansah,

We look forward to receiving your revised manuscript.

Kind regards,

Michele Tizzoni

Academic Editor

PLOS ONE

Reviewers' comments:

Reviewer's Responses to Questions

**Comments to the Author**

Reviewer #1: (No Response)

2. Is the manuscript technically sound, and do the data support the conclusions?

Reviewer #1: Partly

3. Has the statistical analysis been performed appropriately and rigorously?

Reviewer #1: N/A

4. Have the authors made all data underlying the findings in their manuscript fully available?

Reviewer #1: (No Response)

5. Is the manuscript presented in an intelligible fashion and written in standard English?

Reviewer #1: (No Response)

Reviewer #1: The revised version of the paper has addressed many of the key points you previously raised, but some items are handled more thoroughly than others. Below is a summary of how well the updated manuscript responds to the reviewer feedback, keeping in mind the suggestions that were specific to timeliness, data harmonization, threshold development, outcome tracking, and funding sustainability.

First, the authors have clarified the importance of timeliness by incorporating average time intervals. This level of detail had been highlighted as essential, and now the paper supplies numerical data and acknowledges the proportion of samples that met the time standard. The discussion explicitly interprets how delayed care-seeking contributes to lower positivity rates, which aligns with the earlier suggestion to connect patient behavior to detection performance. Although they do not present exact turnaround times for lab feedback beyond stating an average two days once reagents are available, they mention weekly result transmissions and specify the 58% compliance rate for 48-hour sample delivery. That partially satisfies earlier comments.

Threshold development was flagged as a gap before, and the authors now include an account of how they retroactively generated a C-2 threshold for the four-year dataset. This addition underscores that December 2019, April 2021, and July 2021 would have been epidemic signals. They explicitly recognize the absence of a real-time threshold in the district and recommend establishing one. That is a significant improvement over the previous version because it engages with the notion of an “early warning system” more clearly. The paper nevertheless does not describe a formal plan or next steps for implementing thresholds, though it does reference training local staff on using software like Microsoft Excel to plot alerts. This suggests a partial but not comprehensive adoption of the recommendation to institutionalize such a threshold.

Regarding data harmonization, the authors now make a point of explaining discrepancies in the number of suspected cases recorded at different levels. They attribute the mismatch to limited data reconciliation, missing forms at the NIC, insufficient sample volumes, and especially the shift in priority testing during the peak of COVID-19. This acknowledgment indicates they have attempted to address the data harmonization issue, though it remains more of a descriptive explanation rather than a structured solution. The recommendation for routine data reconciliation is briefly included, but it is not expounded upon in detail.

On the topic of tracking patient outcomes and mortality, the authors do confirm that none of the case-based forms contain final outcome data and that mortality is not captured. They cite the need for these fields to be made mandatory but do not detail a formal plan for implementing and monitoring such changes. They do, however, articulate how the lack of outcome data constrains the understanding of disease severity and overall burden, something that the previous review had highlighted as missing. This partial improvement demonstrates awareness of the issue, but further specification on how to close the gap would strengthen this section.

Finally, the paper addresses sustainability and the risk of reliance on donor funds by noting that staff salaries are covered by government sources but that most lab testing supplies and logistical support come from external partners like WHO or the US CDC. This passage shows that the authors have recognized the over-dependency on external donors, which was a previous concern. However, much like in other areas, the revised text identifies the problem but does not systematically propose a sustainability plan. They do suggest greater national budgeting but only briefly. Therefore, they acknowledge the concern more prominently than before, though they stop short of providing a complete blueprint.

The updated version does a better job integrating numerical measures of timeliness, describing data discrepancies, producing a retrospective threshold for outbreak detection, and recognizing the absence of mortality data. It also incorporates the funding and sustainability issues more explicitly. There is still some room to strengthen action steps and outline future plans regarding threshold adoption, routine reconciliation of data, and solutions to reduce reliance on external funding. Nevertheless, overall, the authors have improved the paper by addressing central critiques from the review, making the surveillance evaluation more concrete and actionable.

**Do you want your identity to be public for this peer review?** For information about this choice, including consent withdrawal, please see our Privacy Policy

Reviewer #1: **Yes: ** Onicio Batista Leal Neto

---

## [Author Response · Author response to Decision Letter 2]

17 Jul 2025

Rebuttal letter addressing each point raised by academic editors and reviewers

Major review points:

1. First, the authors have clarified the importance of timeliness by incorporating average time intervals. This level of detail had been highlighted as essential, and now the paper supplies numerical data and acknowledges the proportion of samples that met the time standard. The discussion explicitly interprets how delayed care-seeking contributes to lower positivity rates, which aligns with the earlier suggestion to connect patient behavior to detection performance. Although they do not present exact turnaround times for lab feedback beyond stating an average two days once reagents are available, they mention weekly result transmissions and specify the 58% compliance rate for 48-hour sample delivery. That partially satisfies earlier comments.

Response: This section has been revised or improved regarding turnaround time (TAT) for specimen testing as seen under the discussion section, on line 488 - 493.

“Although our findings indicate an average of 48hours for specimen testing, weekly submission of results and reports to sentinel sites and global platform, system does not capture data on date of specimen testing at the NIC which refutes the calculation of turn-around-time of specimen testing which could impact result feedback for patient care. Data reconciliation as proposed earlier using the SORMAS platform could remedy this gap, where specific date of testing of patients’ specimen could be completed”.

2. Threshold development was flagged as a gap before, and the authors now include an account of how they retroactively generated a C-2 threshold for the four-year dataset. This addition underscores that December 2019, April 2021, and July 2021 would have been epidemic signals. They explicitly recognize the absence of a real-time threshold in the district and recommend establishing one. That is a significant improvement over the previous version because it engages with the notion of an “early warning system” more clearly. The paper nevertheless does not describe a formal plan or next steps for implementing thresholds, though it does reference training local staff on using software like Microsoft Excel to plot alerts. This suggests a partial but not comprehensive adoption of the recommendation to institutionalize such a threshold.

Response: A more comprehensive plan to institutionalize thresholds have been included in discussion section, on line 355 – 362 and to read as;

“A national implementation of threshold development for the ILI surveillance system could be achieved through stakeholder engagement, to review the national influenza protocol for the inclusion of a mandatory development of thresholds, at national, regional, district and sub-district levels. Hence include national level training of all surveillance staff at the lower levels on thresholds development, data entries for its development and interpretations of thresholds. Also, SORMAS (surveillance, Outbreak Response Management and Analysis System), a data platform used by GHS could have its dashboard updated for the calculation of thresholds”.

3. Regarding data harmonization, the authors now make a point of explaining discrepancies in the number of suspected cases recorded at different levels. They attribute the mismatch to limited data reconciliation, missing forms at the NIC, insufficient sample volumes, and especially the shift in priority testing during the peak of COVID-19. This acknowledgment indicates they have attempted to address the data harmonization issue, though it remains more of a descriptive explanation rather than a structured solution. The recommendation for routine data reconciliation is briefly included, but it is not expounded upon in detail.

Response: A detailed expounded data reconciliation has been provided also under discussion section, on line 420 – 444, and this reads as;

“Efforts aimed at curbing data discrepancies in the ILI system could be through an implemented and routine stakeholder quarterly meeting, between both national (GHS and NIC) and district (sentinel site). These routine meetings could be held at the sentinel site where missing information including patients’ outcome, are reconciled, using the source data (i.e., patients’ folders). It also offers an opportunity for the facility to update their forms or data routinely with the results from the NIC lab.

More robustly, the adaptation of SORMAS at the national level which was used during the COVID-19 era could be tweaked to include the capture of influenza data entry interface at all levels as well as at the NIC. The adaptation of SORMAS will foster the mandatory/required completion of all essential data variable on the questionnaire. Through SORMAS, the NIC would be able to update the specimen test results field in real-time for use at the district or health facility level to aid patient care.

To resolve the issue of insufficient sample volumes, the NIC and stakeholders at the national level should periodically train lower-level staff and research assistants that transport samples, on specimen handling. As the NIC continues to supply the Flu site with VTM (viral transport medium), they need to train both their research assistants and the staffs at the sentinel site on the proper cocking of VTMs and thus, safe handling during sample collection and transportation. Whereas COVID-19 samples were prioritized during the pandemic, we will recommend that during the event of outbreaks and pandemics, laboratory testing which are associated with pathogens of the same family could be conducted concurrently”.

4. On the topic of tracking patient outcomes and mortality, the authors do confirm that none of the case-based forms contain final outcome data, and that mortality is not captured. They cite the need for these fields to be made mandatory but do not detail a formal plan for implementing and monitoring such changes. They do, however, articulate how the lack of outcome data constrains the understanding of disease severity and overall burden, something that the previous review had highlighted as missing. This partial improvement demonstrates awareness of the issue, but further specification on how to close the gap would strengthen this section.

Response: This section has been revised and can be found under the discussion section, on line 365 – 371. This reads as follows;

“The ILI surveillance system could improve or integrate data entry into SORMAS at the facility level, which captures the epidemiological data of the patient and thus, be linked to the data entry platform at the NIC for monitoring and subsequent completion of all the required fields after the results are confirmed. Both sides complete all required fields including disease outcomes which foster proper monitoring. This plan implemented by the stakeholders at the national level will help understand disease severity such as mortality, recovery, lost to follow-up, etc.

5. Finally, the paper addresses sustainability and the risk of reliance on donor funds by noting that staff salaries are covered by government sources but that most lab testing supplies and logistical support come from external partners like WHO or the US CDC. This passage shows that the authors have recognized the over-dependency on external donors, which was a previous concern. However, much like in other areas, the revised text identifies the problem but does not systematically propose a sustainability plan. They do suggest greater national budgeting but only briefly. Therefore, they acknowledge the concern more prominently than before, though they stop short of providing a complete blueprint.

Response: This section has been improved with a blueprint to stop or reduce over-dependency on external donors, and can be found under the discussion section, on line 465 – 476. It reads as;

“In as much as the NIC and the influenza surveillance system in Ghana, has over the years depended on funds from external donors, such as the US CDC, WHO and NAMRU-3 long-terms sustainability of the system remains grey. To resolve the overdependency on external donors, sustainability efforts could be through advocacy to highlight the importance of the ILI surveillance system to the government, for the consideration of budget allocation to support equipment procurement/service contracts, reagents and consumables all year round.

During the COVID-19 era, the NIC experienced a novel and heightened private-public partnership including financial institutions, private institutions contributing towards the procurement of equipment, donation of reagents and consumable. The ILI surveillance system could leverage on this cemented private-public partnership, to advocate for financial support as well as grants towards equipment procurement and service contracts, reagents, and consumables for infection-prevention-control (IPC)”.

---

## [Editor Report · Decision Letter 2]

29 Aug 2025

Evaluation of the performance of the Influenza-like Illness (ILI) surveillance system in the Okai Koi North District, Greater Accra Region, 2022

PONE-D-25-02362R2

Dear Dr. Edu-Quansah,

We’re pleased to inform you that your manuscript has been judged scientifically suitable for publication and will be formally accepted for publication once it meets all outstanding technical requirements.

Kind regards,

Michele Tizzoni

Academic Editor

PLOS ONE
---

## [Editor Report · Acceptance letter]

PONE-D-25-02362R2

PLOS ONE

Dear Dr. Edu-Quansah,

I'm pleased to inform you that your manuscript has been deemed suitable for publication in PLOS ONE. Congratulations! Your manuscript is now being handed over to our production team.

Kind regards,

on behalf of

Dr. Michele Tizzoni

Academic Editor

PLOS ONE